# Impact of Perioperative Fluid Strategies on Outcomes in Radical Cystectomy: A Systematic Review

**DOI:** 10.3390/cancers17111746

**Published:** 2025-05-22

**Authors:** Paweł Lipowski, Adam Ostrowski, Jan Adamowicz, Filip Kowalski, Tomasz Drewa, Kajetan Juszczak

**Affiliations:** 1Department of Urology and Andrology, Collegium Medicum, Nicolaus Copernicus University, 85-094 Bydgoszcz, Poland; adostro@gmail.com (A.O.); adamowicz.jz@gmail.com (J.A.); urologpolska@gmail.com (F.K.); t.drewa@wp.pl (T.D.); 2Department of Regenerative Medicine, Collegium Medicum, Nicolaus Copernicus University, 85-094 Bydgoszcz, Poland

**Keywords:** bladder cancer, radical cystectomy, fluid therapy

## Abstract

Fluid management during bladder cancer surgery can affect how much blood a patient loses, how quickly they recover, and whether complications occur. In this review, we looked at different ways of giving fluids during radical cystectomy, including limited (restrictive) fluids, personalized fluid therapy, and using warmed fluids. Some approaches, like limiting fluids and adjusting them based on the patient’s condition, may help reduce blood loss and complications. However, the studies used different surgical techniques and protocols, which makes it hard to say which method works best. Some patients had robotic surgery, others had open surgery, and some had different ways of reconnecting the urinary system. More high-quality research is needed to better understand which fluid therapy is safest and most effective for patients having this major surgery.

## 1. Introduction

Radical cystectomy with urinary diversion remains the standard treatment for muscle-invasive bladder cancer. Despite advancements in surgical techniques and perioperative care, the procedure continues to be associated with significant morbidity and complications [1,2,3,4].

Multiple factors influence postoperative complications following radical cystectomy. Even inflammatory and hematological parameters appear to have an impact on surgical outcomes [5]. Perioperative fluid management plays a crucial role in optimizing postoperative recovery and minimizing complications, including postoperative ileus [6]. Some reports suggest that the total perioperative fluid administration should not exceed 3 L within the perioperative period [7]. While various perioperative fluid management approaches during radical cystectomy have been described in the literature, the optimal strategy remains undefined.

Restrictive fluid therapy is a key component of the Enhanced Recovery After Surgery (ERAS) protocol, which was originally developed for general surgery and has since been successfully implemented in urological oncology, including radical cystectomy (RC). Incorporating ERAS elements such as fluid restriction, early mobilization, and minimized surgical stress has been shown to reduce hospital length of stay and improve recovery outcomes [8,9,10]. Meta-analyses further support its benefits in RC, demonstrating lower complication rates and faster return of bowel function [11].

This systematic review aims to evaluate the impact of different perioperative fluid management strategies on surgical outcomes in patients undergoing radical cystectomy. It focuses on key postoperative complications, including ileus, acute kidney injury, and blood transfusion rates. The review follows the PRISMA guidelines to ensure a comprehensive and unbiased synthesis of the available evidence (Appendix A).

## 2. Materials and Methods

The methodology used in this study adheres to the Preferred Reporting Items for Systematic Reviews and Meta-Analyses (PRISMA) guidelines.

A literature search was conducted in January 2025 using the PubMed database with the following search query:

(“Cystectomy”[Mesh] OR cystectomy OR “RARC” OR “radical cystectomy”) AND (“Perioperative Care”[Mesh] OR “Perioperative Period” OR “ERAS” OR “enhanced recovery after surgery” OR “fluid management” OR “fluid restriction” OR “fluid therapy”) AND (“Postoperative Complications”[Mesh] OR complications OR “acute kidney injury” OR “acute kidney failure” OR “hospital stay” OR “bowel function” OR “analgesic use” OR pain OR readmission OR mortality OR “quality of life” OR “QoL” OR “life quality” OR “surgical site infection” OR “SSI” OR ileus OR “blood transfusion” OR “electrolyte imbalance”) AND (“Urinary Bladder Neoplasms”[Mesh] OR “bladder cancer” OR “bladder neoplasm”). The retrieved studies were subsequently analyzed.

Randomized controlled trials (RCTs) and observational studies published in English, evaluating the impact of different perioperative fluid therapy strategies during radical cystectomy on postoperative outcomes and complications, were included in the review.

The selection process was conducted by two independent reviewers and verified by a third researcher who resolved any discrepancies. Studies without full-text availability were excluded. The selection process is illustrated in the PRISMA flowchart (Figure 1).

Data were extracted regarding authors, year of publication, study type, study population, intervention type, and key outcomes. Study quality was assessed using the Cochrane Risk of Bias Tool (version RoB2). Risk of bias was assessed using the Cochrane Risk of Bias Tool, evaluating domains such as sequence generation, allocation concealment, blinding, completeness of outcome data, and selective reporting. Each domain was classified as having low, high, or unclear risk of bias based on the criteria outlined in the *Cochrane Handbook*. A narrative synthesis was performed to evaluate significant differences in study designs.

Reporting bias (e.g., publication bias or selective outcome reporting) was not formally assessed due to the narrative nature of the synthesis and the limited number of studies per outcome. Effect measures included mean differences for continuous outcomes (e.g., blood loss and length of stay) and proportions or risk differences for binary outcomes (e.g., complications, transfusions, and AKI). Where available, *p*-values were reported. Other effect measures, such as odds ratios or confidence intervals, were not consistently provided across studies.

The certainty of evidence for selected key outcomes (acute kidney injury, blood loss and transfusions, postoperative ileus, length of hospital stay, and chronic kidney disease) was assessed using an adapted GRADE approach, taking into account risk of bias, consistency, precision, and directness across studies.

This review was not prospectively registered.

## 3. Results

### 3.1. Study Selection and Characteristics

A total of 538 articles were identified in the initial search. After screening, 31 articles were subjected to full-text evaluation. Following a rigorous selection process, 17 articles were included in the systematic review.

These 17 studies focused on comparing perioperative fluid therapy strategies in patients undergoing radical cystectomy with lymphadenectomy and urinary diversion for bladder cancer. Among them, 8 studies were randomized controlled trials (RCTs) with a double-blind design, including 3 articles analyzing the same population, with 1 being a follow-up study. The remaining 9 studies were observational, including 6 retrospective studies, 1 prospective study, and 2 studies comparing retrospective and prospective cohorts.

Regarding the surgical approach, 8 studies analyzed patients undergoing open radical cystectomy (ORC), 1 study assessed patients treated with robot-assisted laparoscopic radical cystectomy (RARC), and 1 study included both RARC and ORC. In 5 studies, the surgical approach was not explicitly defined. Additionally, 8 of the included studies incorporated the ERAS (Enhanced Recovery After Surgery) protocol.

The total patient population analyzed across all included studies comprised 3519 patients. The characteristics of the included studies are presented in Table 1. The risk of bias for the included trials was assessed and is summarized in Table 2, where variability in methodological quality is highlighted as a factor potentially influencing the strength of the evidence. Potential reporting bias was not evaluated.

### 3.2. Restrictive Fluid Therapy with Norepinephrine Use

The use of norepinephrine infusion combined with restrictive fluid therapy is described in three publications based on a single randomized clinical trial [20,21,22].

A total of 83 patients were included in the study group, and 84 in the control group. The results indicate that the application of restrictive fluid therapy combined with norepinephrine infusion significantly reduces blood loss (800 mL vs. 1200 mL; *p* < 0.0001) and decreases the need for blood product transfusion (33% vs. 60%; *p* = 0.0006). This approach also lowers the risk of complications during hospitalization (52% vs. 73%; *p* = 0.006), including gastrointestinal complications (6% vs. 31%; *p* < 0.0001) and cardiovascular complications (20% vs. 48% *p* = 0.0003). Moreover, it shortens the length of hospital stay (15 days vs. 17 days; *p* = 0.02) and reduces mortality within 90 days postoperatively (0% vs. 4.8%; *p* = 0.12).

### 3.3. Goal-Directed Fluid Therapy (GDFT)

Goal-directed fluid therapy refers to individualized fluid administration guided by dynamic hemodynamic parameters to optimize tissue perfusion and avoid fluid overload. The use of GDFT is described in six studies, including four randomized trials [9,10,11,13] and two observational studies [17,28]. The total number of patients in the study groups was 470, while 551 patients were assigned to the control groups. Both observational studies incorporated the ERAS protocol.

Ghoreifi et al. [17] found no significant differences between GDFT and conventional fluid therapy in terms of blood loss (567 mL vs. 580 mL; *p* = 0.6), eGFR trend (*p* = 0.85), length of stay (4 days in both groups; *p* = 0.85), 90-day high-grade complication rate (17.6% vs. 17.2%; *p* = 0.5), or 90-day readmissions (28.6% vs. 29.7%; *p* = 0.9). Patel et al. [28] reported a lower intraoperative blood transfusion rate (0.58 vs. 0.97; *p* ≤ 0.001) and a lower incidence of nausea (3.1% vs. 8.1%; *p* < 0.05) in the GDFT group. However, no significant differences were observed in length of stay (LOS) (6 days in both groups; *p* = 0.89), peak pain levels (4.4 vs. 4.4; *p* = 0.89), or 30- and 90-day readmissions (*p* = 0.34 and *p* = 0.14, respectively). The authors noted no differences in time to first flatus or 30- and 90-day complication rates, but *p*-values were not provided for these variables.

Randomized trials assessing GDFT using Doppler ultrasound demonstrated a lower risk of ileus (22% vs. 53%; *p* < 0.001), nausea (9% vs. 32% *p* < 0.01), or time to flatus (3.55 days vs. 5.36 days; *p* < 0.01 [15]. Wound infection rates were reduced in the study group (1 vs. 8; *p* < 0.01). Liu et al. [16] demonstrated a lower risk of nausea, vomiting, and hypotension (all *p* < 0.05). Patients in the intervention groups experienced a faster return of bowel function (2.1 days vs. 3.0 days; *p* < 0.05); however, the length of hospital stay was shorter in the control group (11.5 days vs. 13.7 days; *p* < 0.05).

GDFT based on stroke volume variation (SVV) was associated with reduced blood loss (734.3 mL vs. 1096.5 mL; *p* = 0.019) and a lower transfusion rate (0.5 units vs. 1.9 units; *p* = 0.005). However, this did not significantly affect gastrointestinal function (*p* = 0.326, *p* = 0.4) or length of hospital stay (*p* = 0.23, *p* = 0.9) [13].

The data regarding acute kidney injury (AKI) are conflicting. A study with a larger sample size reported a higher risk of AKI in the GDFT group (56% vs. 40%; *p* = 0.005) [14].

Given the variation in study designs, populations, interventions, and outcome measures, a quantitative synthesis was deemed inappropriate.

### 3.4. Warming Fluid Transfusion

A randomized clinical trial by Luo et al. indicates that perioperative fluid therapy using warmed fluids at 38–42 °C in patients undergoing robot-assisted radical cystectomy (RARC) reduces fluid transfusions (1903 mL vs. 2153 mL; *p* = 0.028) and shortens the length of hospital stay (16 days vs. 20 days; *p* = 0.05). However, it does not affect blood loss, the need for reoperation, or bowel function [12].

### 3.5. Vascular Bed Filling Index

In another study of ours, we proposed the use of the Vascular Bed Filling Index (VBFI) and the adjusted Vascular Bed Filling Index (aVBFI) to determine the optimal volume of perioperatively administered fluids [23]. These indices are based on blood loss, the volume of perioperatively administered fluids, and the duration of the surgical procedure. The results suggest that fluid therapy guided by these indices may reduce the severity of complications after radical cystectomy, particularly in patients undergoing urinary diversion using an ileal conduit (VBFI = between 0 and 8; *p* = 0.011 for VBFI; *p* = 0.005 for aVBFI). Although the VBFI/aVBFI-guided approach showed a reduction in complications in patients undergoing ileal conduit diversion, it should be considered exploratory and hypothesis-generating, as it is based on a single retrospective study from the authors’ institution.

### 3.6. Anesthetist Impact

A retrospective study evaluated the impact of anesthesiologist experience on the outcomes of radical cystectomy. The study indicates that greater anesthesiologist experience (>10 cases) is associated with reduced blood loss (600 mL vs. 800 mL, *p* < 0.001) and a lower transfusion rate (7.2% vs. 22%, *p* = 0.001). However, it does not affect mortality or the need for rehospitalization [27].

### 3.7. Impact on Specific Complications

#### 3.7.1. Acute Kidney Injury

Based on assessments of four studies (included two randomized trials and two retrospective observational studies), the relationship between perioperative fluid therapy and acute kidney injury (AKI) was analyzed. Both randomized trials compared the effects of goal-directed fluid therapy (GDFT) with control groups.

In the randomized studies, a total of 165 patients were included in the intervention groups, and 164 patients in the control groups. In both studies, the risk of AKI was higher in the intervention group. In a study by Kong et al. [13], 8 patients (34%) in the GDFT group and 6 patients (26%) in the control group developed AKI (*p* = 0.522). However, these numbers did not reach statistical significance, likely due to the small sample size. In contrast, Arslan-Carlon et al. [14] analyzed a larger cohort, where the risk of AKI was significantly higher in the restrictive fluid therapy group (56% vs. 40%, *p* = 0.005).

The results of observational studies are inconsistent. One observational study also indicates a higher risk of AKI in patients who received <5 mL/kg/h of perioperative fluids (*p* = 0.043, Citation 15). Conversely, Furrer et al. suggests that the administration of >2527 mL of crystalloids intraoperatively is associated with an increased risk of AKI (*p* = 0.3) [22].

#### 3.7.2. Blood Loss and Transfusions

The results of four randomized clinical trials assessed perioperative blood loss and transfusion rates. Two studies investigated goal-directed fluid therapy (GDFT), one examined warming fluid transfusion, and one evaluated norepinephrine-assisted restrictive fluid therapy.

Restrictive fluid therapy with norepinephrine significantly reduced intraoperative transfusion rates (8% vs. 31%; *p* < 0.001) and postoperative transfusion requirements (28% vs. 48%, *p* = 0.0006) [25].

The findings regarding GDFT were inconsistent. Kong et al. [15] demonstrated that stroke volume variation (SVV) monitoring for fluid optimization significantly reduced blood loss (734 mL vs. 1096 mL; *p* < 0.001) and the number of transfusions (0.5–0.8 units vs. 1.9–2.2 units; *p* = 0.005) [13]. Conversely, Pillai et al. [15] found that Doppler ultrasound-based fluid therapy had no significant impact on blood loss (*p* = 0.6) or red blood cell transfusion requirements (*p* = 0.82). However, both of these studies had small sample sizes, with 23 vs. 23 and 32 vs. 34 patients in the study and control groups, respectively.

Additionally, warming fluid transfusion at 43 °C during cystectomy did not significantly affect blood loss (*p* = 0.19) [12].

#### 3.7.3. Postoperative Ileus

Postoperative ileus is a significant concern in patients undergoing radical cystectomy. This issue was addressed in four clinical studies, with a total of 334 patients in the intervention groups and 337 patients in the control groups.

The use of low-dose norepinephrine was associated with a significant reduction in the risk of ileus (6% vs. 37%; *p* = 0.007) and gastrointestinal complications (6% vs. 31%; *p* < 0.0001) [24].

The remaining findings regarding various GDFT strategies were inconsistent. Most data indicate no significant effect of GDFT on the incidence of postoperative ileus (*p* = 0.4–(14); *p* = 0.3–(13)).

Only Pillai et al. [15] demonstrated a positive impact of GDFT on postoperative ileus (*p* < 0.001), bowel motility (*p* = 0.01), and time to first bowel movement (*p* = 0.02); however, this study had a small sample size.

### 3.8. Length of Stay

The findings regarding hospital length of stay (LOS) are as follows: Restrictive fluid therapy combined with norepinephrine infusion significantly reduced hospital stay (15 days vs. 17 days; *p* = 0.01) [24]. In contrast, GDFT did not impact LOS (7 days vs. 7 days; *p* = 0.9) [14], while warming fluid transfusion at 38–42 °C was associated with a shorter hospital stay (16 days vs. 20 days; *p* = 0.05) [12].

All of the above findings are derived from randomized clinical trials; however, it is noteworthy that significant differences were observed only in studies where the average hospital stay exceeded several days.

Additionally, an observational study using SVV-based GDFT found no difference in LOS between groups (4 days vs. 4 days; *p* = 0.85) [17].

### 3.9. Chronic Kidney Disease

A randomized clinical trial comparing conventional fluid therapy with restrictive fluid therapy combined with norepinephrine infusion found no significant differences in kidney function between groups after 7 days (*p* = 0.37) and at 3, 6, and 12 months after discharge (*p* = 0.49; *p* = 0.197; and *p* = 0.78, respectively) [26].

However, the study identified independent risk factors for kidney function deterioration, including diabetes (*p* = 0.002), preoperative eGFR (*p* = 0.007), and age (*p* = 0.038).

### 3.10. Certainty of Evidence (GRADE Assessment)

The overall certainty of evidence for outcomes was low to moderate. Certainty ratings were based on study design, consistency of results, precision of estimates, and risk of bias, following the GRADE framework. According to the GRADE assessment, the evidence was rated as low for acute kidney injury, postoperative ileus, and length of stay due to inconsistency and imprecision across studies. Moderate certainty was assigned to outcomes such as blood loss and transfusions and chronic kidney disease based on consistent findings and lower risk of bias. Detailed ratings are presented in Table 3.

## 4. Discussion

This systematic review assessed the impact of various perioperative fluid therapy strategies in patients undergoing radical cystectomy with lymphadenectomy and urinary diversion due to bladder cancer. The analyzed studies exhibit significant heterogeneity, and some contain methodological limitations, making direct comparisons between studies challenging. It is important to note that radical cystectomy and its postoperative complications vary significantly depending on the type of urinary diversion used [29]. This factor further complicates direct comparisons between studies.

The results of the current systematic review suggest that the most optimal approach involves restrictive fluid therapy combined with low-dose norepinephrine infusion or goal-directed fluid therapy (GDFT) using stroke volume variation (SVV) for fluid administration. The administration of pre-warmed fluids also appears to be a noteworthy strategy.

However, it is essential to consider that the Enhanced Recovery After Surgery (ERAS) protocol is actively implemented during radical cystectomy, contributing to reduced postoperative complications and faster recovery [26,27,28,29,30]. Since all components of the ERAS protocol work synergistically, evaluating fluid therapy in isolation from the protocol may be controversial. In this review, the ERAS protocol was applied in eight studies, but the methodologies of these studies do not allow for direct comparisons.

Restrictive fluid therapy with norepinephrine during radical cystectomy appears to be a promising strategy; however, ERAS was not implemented in these studies [20,21,22]. This approach has also been investigated in other surgical fields, including liver surgery [31], thoracic surgery [32], and head and neck oncological surgery [33], showing favorable outcomes.

The GDFT strategy, based on stroke volume variation or transesophageal Doppler monitoring, also appears to be a beneficial approach. However, randomized controlled trials evaluating these techniques included small sample sizes and did not involve concurrent ERAS protocol implementation [9,11,12].

Fluid therapy with pre-warmed fluids at approximately 41 °C may also be a valuable strategy. Although the total population of studies investigating this method during radical cystectomy is relatively small, this approach has demonstrated efficacy in other surgical procedures, including in elderly patients [33,34,35,36].

Differences in surgical approach (robotic vs. open) and urinary diversion technique (intracorporeal vs. extracorporeal) may influence key perioperative outcomes such as bleeding, postoperative ileus, and length of stay, and should therefore be taken into account when evaluating the effects of fluid management strategies [37,38].

The studies included in this review also differ in terms of surgical technique. It is important to emphasize that a significant proportion of radical cystectomies are now performed using laparoscopic or robot-assisted techniques. Robot-assisted surgery is associated with a longer operative time but a lower need for blood transfusion [37,38]. Based on these findings, the optimal fluid therapy strategy should also be considered in conjunction with the surgical approach.

The use of the Vascular Bed Filling Index and adjusted Vascular Bed Filling Index is based on a single retrospective study and requires further research to draw definitive conclusions.

This review was not prospectively registered, and only one database (PubMed) was searched, which may have led to selection bias. Only English-language publications were included, potentially omitting relevant evidence. Although study selection was conducted by two independent reviewers, data extraction and bias assessment were not independently duplicated. Reporting bias was not assessed, and certainty of evidence was evaluated only for selected outcomes. The included studies exhibited substantial heterogeneity in terms of design, population, surgical approach, and outcome reporting. A significant proportion of studies were observational and retrospective, with potential confounding and variable implementation of ERAS protocols. Additionally, the lack of standardized definitions for key outcomes, such as postoperative ileus or complications, limits comparability across studies. The presence of high or unclear risk of bias in multiple studies highlights the need for cautious interpretation of the findings and underlines the importance of high-quality randomized trials in this field.

Based on current evidence, restrictive fluid therapy with norepinephrine may reduce perioperative complications and blood loss, and could be considered as part of perioperative strategies in radical cystectomy, especially where ERAS protocols are not fully implemented. GDFT shows potential for reducing gastrointestinal complications and improving recovery, although findings on renal outcomes are inconsistent and warrant caution.

Future research should focus on direct comparisons of fluid therapy strategies in conjunction with the ERAS protocol and modern surgical techniques, with particular emphasis on the type of urinary diversion used.

## 5. Conclusions

Restrictive therapy with norepinephrine reduces blood loss and complications, while goal-directed fluid therapy improves hemodynamics but has mixed effects on complications and hospital stay. Warming fluids lower transfusions, and Vascular Bed Filling Index–based management may reduce complications, though evidence is limited. Further research is needed to determine the optimal approach.

## Figures and Tables

**Figure 1 cancers-17-01746-f001:**
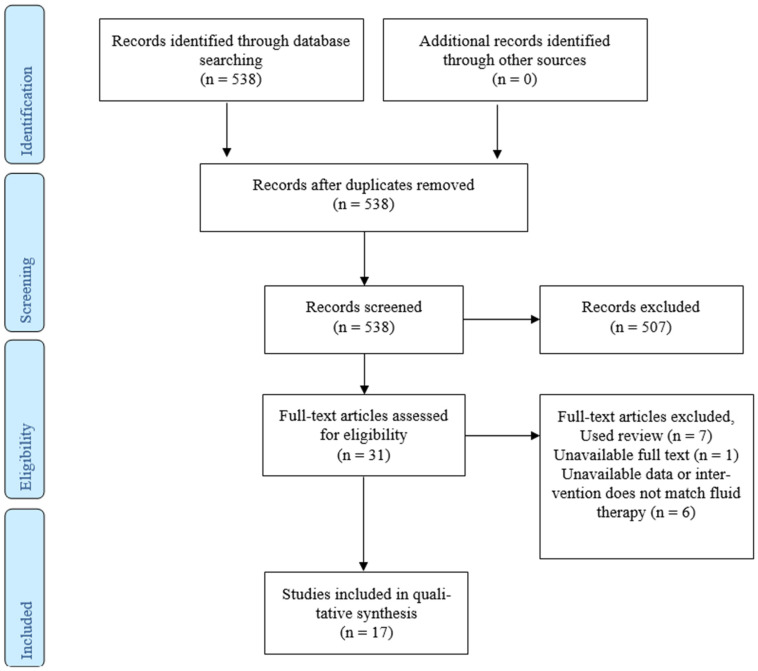
PRISMA flowchart of the study selection process.

**Table 1 cancers-17-01746-t001:** Characteristics of included studies.

No.	Author (Year)	Study Design	Participants (Study Group vs. Control Group)	Intervention	Outcomes
1	Luo J. (2020) [12]	RCT	53 vs. 55	Pre-warm fluid infusion use	↓ transfusion, ↓ LOS
2	Kong Y. (2016) [13]	RCT	23 vs. 23	Fluid infusion based on stroke volume variation (SVV) 10–20% vs. <10%	↓ transfusion, ↓ blood loss. ↔ complications, ↔ AKI, ↔ LOS
3	Arslan-Carlon V. (2020) [14]	RCT	142 vs. 141	Goal-directed stroke volume (SV) vs. standard fluid therapy	↑ 30-day complications, ↑ acute kidney injury ↔ ileus, ↔LOS
4	Pillai P. (2011) [15]	RCT	32 vs. 34	Doppler-guided vs. standard fluid therapy	↓ ileus, ↓ nausea, ↓ vomiting. ↓wound infection.
5	Liu T. (2016) [16]	RCT	38 vs. 38	Goal-directed SVV vs. routine fluid therapy	↑ cardiac output index, ↑ MAP, ↑ central venous pressure. Better metabolic index. ↓ nausea, ↓ vomiting, ↓ hypotension.
6	Ghoreifi A. (2021) [17]	Retrospective cohort	119 vs. 192	Goal-directed SVV vs. convectional fluid therapy	↔ kidney function, ↔ LOS, ↔ blood loss, ↔ transfusions, and readmissions.
7	Wei C. (2018) [18]	Retrospective cohort	91 vs. 101	ERAS (less fluids) vs. no ERAS	↓ blood loss, ↓ transfusions, ↓ readmissions, and ↓ complications. ↓ bowel complications.
8	Bazargani T. (2018) [19]	Prospective cohort	180	Total intraoperative fluid volume, type of fluid impact	↔ 30 -, 90-day complications, ↔ in LOS.
9	Dobe T. (2022) [20]	Prospective vs. retrospective control group	29 vs. 50	ERAS (less fluids + vasopressive drugs) vs. no ERAS	↔ blood loss, ↔ transfusions, ↔ ileus, ↔ complications rate. ↓ LOS.
10	Marques M. (2024) [21]	Retrospective cohort	51 vs. 71	No AKI vs. AKI	↑ AKI if restrictive intraoperative vascular filling, female sex, postoperative sepsis, day 1 SOFA score, creatinin D1.
11	Furrer M. (2018) [22]	Retrospective cohort	100 vs. 812	AKI vs. no AKI	↑ AKI if surgery time >400 min, male, obesity, high blood loss, blood transfusion, more crystalloids.
12	Lipowski P. (2024) [23]	Retrospective cohort	48 vs. 240	Ileal conduit (IC) vs. ureterocutaneostomy (UCS), Vascular Bed Filling Index (VBFI), adjusted Vascular Bed Filling Index (aVBFI)	VBFI, aVBFI: < 8–UCS ↓ complications; = 8–IC = UCS; >8–IC ↑ complications.
13	Wuethrich P. (2013) [24]	RCT	83 vs. 84	Low volume + noradrenaline vs. balanced Ringer’s solution	↓ hospital complications, ↓ gastrointestinal, ↓ cardiac, ↓ 90 days complications. ↓ LOS.
14	Wuethrich P. (2013) [25]	RCT	83 vs. 84	Low volume + noradrenaline vs. balanced Ringer’s solution	↓ blood loss, ↓ transfusions.
15	Mei Wen Wu F. (2013) [26]	RCT	83 vs. 84	Low volume + noradrenaline vs. balanced Ringer’s solution	↔ in renal function 7 days, 3, 6, and 12 months after surgery.
16	Jubber I. (2019) [27]	Retrospective cohort	430	High-volume anesthetist vs. low-volume anesthetist	↓ LOS, ↓ blood loss, ↓ transfusion rate.
17	Patel S. (2018) [28]	Retrospective cohort	116 vs. 143	Multidisciplinary ERAS (goal-directed fluid therapy) vs. surgical ERAS	↓ intraoperative transfusions, ↓ nausea. ↔ in bowel function, ↔ LOS, ↔ 30 and 90 days complications, ↔ readmissions.

Abbreviations: LOS—length of stay; AKI—acute kidney injury; VBFI—Vascular Bed Filling Index; aVBFI—adjusted Vascular Bed Filling Index; ERAS—Enhanced Recovery After Surgery; ↓—significant reduction (favorable effect); ↑—significant increase (unfavorable effect); ↔—no significant difference.

**Table 2 cancers-17-01746-t002:** Cochrane Risk of Bias (risk (high/low/unclear).

No.	Author (Year)	Random Sequence Generation	Allocation Concealment	Blinding of Participants and Personnel	Blinding of Outcome Assessment	Incomplete Data	Selective Reporting	Main Limitations/Quality Considerations
1	Luo J. (2020) [12]	Unclear	Unclear	Unclear	Unclear	Low	Low	Randomization and blinding procedures poorly described, potential risk of bias.
2	Kong Y. (2016) [13]	Low	Low	Low	Low	Low	Low	Good methodology but small sample size (risk of imprecise estimates).
3	Arslan-Carlon V. (2020) [14]	Low	Low	Low	Low	Low	Low	Good methodology, large sample size, low risk of bias.
4	Pillai P. (2011) [15]	High	Unclear	Unclear	Unclear	Low	Low	Small sample size, unclear, randomization and blinding procedures, high risk of bias.
5	Liu T. (2016) [16]	High	Unclear	Unclear	Unclear	Low	Low	Small sample size, unclear randomization details, moderate risk of bias.
6	Ghoreifi A. (2021) [17]	High	High	High	High	Low	Low	Retrospective observational design, possible selection and confounding biases.
7	Wei C. (2018) [18]	High	High	High	High	Low	Unclear	Retrospective, observational study design, risk of selection and confounding biases.
8	Bazargani T. (2018) [19]	High	High	High	High	Low	Low	Observational design without clear control group, potential confounding bias.
9	Dobe T. (2022) [20]	High	High	High	High	Low	Low	Small sample size, mixed prospective-retrospective groups, risk of confounding and selection bias.
10	Marques M. (2024) [21]	High	High	High	High	Low	Low	Retrospective design, limited to AKI risk factors, potential confounding bias.
11	Furrer M. (2018) [22]	High	High	High	High	Low	Low	Retrospective design, large population but significant confounding risks.
12	Lipowski P. (2024) [23]	High	High	High	High	Low	Low	Single-center, retrospective, exploratory and hypothesis generating.
13	Wuethrich P. (2013) [24]	Low	Low	Low	Low	Low	Low	Strong methodology, single population, limited external validity.
14	Wuethrich P. (2013) [25]	Low	Low	Low	Low	Low	Low	Strong methodology, single population, limited external validity
15	Mei Wen Wu F. (2013) [26]	Low	Low	Low	Low	Low	Low	Strong methodology, single population, limited external validity
16	Jubber I. (2019) [27]	High	High	High	High	Unclear	Low	Retrospective design, high-volume anesthetist group limited to few providers, risk of bias.
17	Patel S. (2018) [28]	High	High	High	High	Low	Low	Retrospective design, historical control group, significant selection and confounding biases.

**Table 3 cancers-17-01746-t003:** Grade summary table.

Outcome	No. of Studies	Consistency	Precision	Risk of Bias	Certainty (GRADE)
Acute Kidney Injury (AKI)	4 (2 RCTs, 2 obs.)	Inconsistent	Imprecise (small N, some *p* > 0.05)	Moderate to high (obs. studies, limited blinding)	Low
Blood Loss and Transfusions	4 RCTs	Partially consistent	Moderate (some large effects)	Low to moderate (RCTs well reported)	Moderate
Postoperative Ileus	4 studies (mixed)	Mixed results	Low (small N in key studies)	High in some studies	Low
Length of Hospital Stay	4 RCTs + obs.	Inconsistent	Low (heterogeneity of LOS reporting)	Moderate	Low
Chronic Kidney Disease	1 RCT	Consistent	Moderate	Low	Moderate

Reporting bias was not formally evaluated, and no results related to publication bias are available.

## Data Availability

Data sharing is not applicable to this article as no new data were created or analyzed in this study.

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
