# Peer review of "Impact of Perioperative Fluid Strategies on Outcomes in Radical Cystectomy: A Systematic Review"

_cancers, 2025, doi:10.3390/cancers17111746_

Round 1
Reviewer 1 Report
Comments and Suggestions for Authors
This systematic review focuses on different strategies for perioperative fluid management in patients undergoing radical cystectomy, and the topic is very clinically significant. The authors strictly followed the PRISMA 2020 guidelines, systematically searched and included multiple randomized controlled trials (RCTs) and observational studies, and standardized the literature screening and data extraction process, which enhanced the reliability of the results.
The article is well organized in the results section, summarizing the effects of restrictive fluid therapy, goal-directed fluid therapy (GDFT), and warm fluid application on different perioperative outcomes (blood loss, transfusion rate, complications, length of hospital stay, etc.), and can frankly point out the heterogeneity and inconsistency in different research results, showing the author's ability to critically analyze the literature.
The conclusion section can accurately combine the existing evidence and propose that restrictive fluid therapy combined with norepinephrine may have the most potential, but also points out that GDFT and warm fluid therapy still need more high-quality research verification, reflecting a rigorous and prudent attitude.
Overall, this review has a novel topic, clear logic, and comprehensive literature coverage, and has high clinical reference value and publication potential.
Suggestions:
If possible, the discussion of the differences in the quality of each study can be further strengthened in the full text, such as adding a "risk bias" assessment table or briefly summarizing the main limitations of each study to enhance the depth of the systematic review.
When describing the heterogeneity of the studies, a "qualitative comparison table" (such as a table that summarizes the basic characteristics and conclusions of the main studies) can be appropriately introduced to make the reading experience more intuitive and smooth.
Author Response
Dear reviewer, The authors of the manuscript would like to thank for positive all critical comments and suggestions that are useful for improving the current draft. Ours reply can you find below in added file. Best regards, Paweł Lipowski
Reviewer 2 Report
Comments and Suggestions for Authors
The authors present a meta-analysis of fluid replacement therapies utilized during radical cystectomy. A very important consideration when completing an analysis like this is to identify the most clinically important endpoint to compare across the different replacement strategies. After this is completed, only include studies in the primary analysis that evaluate this endpoint. The selected endpoint should be as objective as possible. I feel the most appropriate endpoint to consider in this population is transfusion rate. This is based on the association with transfusion and survival outcomes and it should be easy to capture and compare between studies. Other endpoints such as AKI, complications, and ileus can be very subjective and likely not recorded consistently between studies.
With this in mind. I would complete a primary analysis of studies comparing estimated blood loss and transfusion rates. It is important to compare EBL as the threshold for blood transfusions could vary greatly between institutions.
Author Response

(The authors gave the same response as above.)

Reviewer 3 Report
Comments and Suggestions for Authors
title - clearly depicting the focus of the study - No remarks
Abstract - row 34 - abbreviation VBFI has not been explained in first usage - Minor
Introduction - insufficient - should be expanded, at least with a paragraph on ERAS - Major
Material and methods - comprehensive description of the study protocol - No Remarks
Results - Table 1 - formatting issues - e.g. outcomes are merging - Major
Discussion and conclusion - narrative presentation of the results observed in the studies included in the analysis - No remarks
Author Response

(The authors gave the same response as above.)

Reviewer 4 Report
Comments and Suggestions for Authors
This review is clinically relevant and well written, with a very interesting subject.
However, several areas require clarification.VBFI/aVBFI strategy is presented by the authors as promising, but the evidence cited is limited to a single, retrospective study from the authors' own group. I think it should be clearly described as exploratory and hypothesis-generating, thus increasing transparency and avoiding being misleading for the readers.
Also, please explain how the ratings in the risk of bias table (table 2) were assigned. Also, Table 2 is not referenced in the text. As this is an important element of systematic review methodology, it should be clearly introduced in the Methods or Results sections and, maybe, briefly discussed in terms of its influence on the strength of the evidence.
Author Response

(The authors gave the same response as above.)

Round 2
Reviewer 2 Report
Comments and Suggestions for Authors
The authors did not directly address my review, it was essentially dismissed.
I will defer to the editor if the manuscript is worth publishing based on endpoints selected by the authors
Reviewer 3 Report
Comments and Suggestions for Authors
the authors had addressed this reviewer`s comments sufficiently. The manuscript is acceptable for publication
Reviewer 4 Report
Comments and Suggestions for Authors
The authors addressed all the queries appropriately. I believe that the article is its current form is suitable to be accepted for publication